# Seasonality of Carabid Beetles on an Organic Agricultural Field and Its Effect on Foraging Use

Ulrich Irmler

Institute of Ecosystem Research, University of Kiel, Olshausenstrasse 40, 24118 Kiel, Germany; uirmler@ecology.uni-kiel.de

**Abstract:** Ground beetle species from marginal areas invade organically farmed fields in a higher abundance and species richness than conventionally farmed fields. Seasonal invasion into organic fields was studied at Ritzerau Manor, converted to organic farming 18 years ago. Carabid species were explored with 123 pitfall traps within the field and in marginal near-natural habitats over the 18 years after conversion. For 56 species, seasonality could be studied in a distance gradient from the field margin to the field center. The results revealed that ground beetles from marginal habitats can use the fields differently depending on their seasonal activity. Early and fast-moving species can reach the center of the field at a 240 m distance from margin; late and slowly moving species only reach the 120 to 60 m distance level. The foraging effect of species, thus, depends on the seasonality and duration of activity. Overall, marginal species make up to 35% of the total foraging of ground beetles. Thus, organic farming not only supports a closer interaction between farmland and the adjacent near-natural landscape, but also benefits from higher biological pest control by immigrating marginal species.

**Keywords:** seasonality; organic land-use; dispersion speed; foraging effect; ground beetles



## 1. Introduction

The biodiversity crisis is a worldwide phenomenon [1]. Agricultural use is one of the most important reasons for species extinction, e.g., by the clear cutting of forests or degradation of soils [2,3]. In Central Europe, the intensification of agriculture is one of the major causes for the extinction of species and the overall decrease in biodiversity [4,5]. The increase in organic agriculture in landscape management is one of the solutions to prevent further diversity degradation and to promote biodiversity [6]. These facts provoked Günther Fielmann, the owner of the fields under study, to change from intensive agricultural management on his farms to organic farming practices.

On one of his farms, Ritzerau Manor, he initiated scientific research in 2001 for the long-term monitoring of the development of biocenosis and biodiversity after the shift to organic management. Since then, nearly 20 years of data have been recorded for various organisms. These long-term trends have already been analyzed in numerous publications [7–11]. The long-term immigration and emigration processes of carabids have also been published in detail [12]. The results show that after 10 years, biodiversity has increased mainly from the margins [9]. The invading species have originated from open habitats, whereas silvicolous species have retreated [7,8]. Additionally, the earthworm fauna has changed with the increasing density of anecic species, also indicating changes in the water-balance of the soil [10]. Whereas the long-term changes over the period of the first 15 years are now well known, the short-term effects during seasons were not analyzed. The present paper will fill this gap and focus on the changes caused by seasonal migration processes, and the combined effects on the foraging use of the area by carabid beetles. Thus, two questions have to be answered: (1) How fast is seasonal immigration into the field, and (2) what effect does the seasonal immigration of species have on foraging use?

## 2. Methods and Sites

The investigations were executed on the arable fields and adjacent field margins of "Hof Ritzerau", located in southeastern Schleswig-Holstein, Germany, near the city of Mölln. The farm was the property of the city of Lübeck for many years, and inherited by Günther Fielmann in 1998. He changed the management of the land from intensive conventional farming to organic farming according to the guidelines of the "Bioland" rules. The change took place from 2001 to 2003. Since 2004, organic farming has been practiced on the entire field area. The research started in 2001 with an intensive phase, and lasted until 2004. A subsequent monitoring phase followed, which is still ongoing. The 290-ha farm is part of a diverse landscape with forests, wet grassland, ponds, and creeks. The study area is comprised of 180 ha of arable fields and approximately 18 ha of near-natural habitats with shrubs, hedges, and ponds that are assigned as adjacent field margins.

A comparison of the investigated Ritzerau fields with 53 other field sites in Schleswig-Holstein showed that the overall differences in carabid assemblages represent weak independent farming practices [7]. Nevertheless, fields can be separated into loamy and extreme sandy sites. The investigated Ritzerau fields belong to the group on loamy sites, which is most common in Schleswig-Holstein. Thus, the investigated landscape can be considered as representative for most of the arable field situations in Schleswig-Holstein.

The climate of the area is characterized by the transitional Atlantic type with a mean yearly temperature (30 years mean) of 9.39 °C and a yearly rainfall range from 487 to 970 mm. Within the study period from 2001 to 2018, no significant increase in temperature was determined. Nevertheless, a long-term trend exists in the period from 1950 to present with an increase of +1 °C and an increase of rainfall from 576 mm $y^{-1}$ to 737 mm $y^{-1}$.

The investigations started in May 2001. Crop rotation under the intensive period from 2001 to 2003 was composed of corn, winter wheat, and winter rape; crop rotation under organic practices since 2004 has been composed of winter wheat, summer wheat, winter rye, summer barley, peas, and a grass-clover mixture. From time-to-time, parts of the fields were used as sheep pasture.

Ground beetles have been recorded since 2001 in the arable fields and the adjacent field margins including hedges, shrubs, and grassland. Over the total study period, a total of 96 pitfall traps were installed on the arable fields and 27 pitfall traps on the near-natural habitats. The pitfall traps on the fields were installed in a grid pattern (Figure 1). The location of the pitfall traps was defined by GPS to verify the identical location throughout the years. From 2002 to 2004, a yearly investigation was performed. During the monitoring phase since 2004, pitfall traps were only installed every second year. Pitfall traps were made from commercially available glass jars with an opening of 5.7 cm diameter and 11 cm height. They were filled with monoethylene–glycol. Transparent plastic shelters were installed against rainfall. Pitfall traps were changed every month throughout the whole year. They were removed in August and parts of September due to harvesting and the sowing of new crop seed. In cold, snow-rich winters, a monthly change of the pitfall traps was sometimes not possible. For the present study, the data from 2002 to 2018 was used. The year 2001 was omitted because data was not available for the whole year. Additionally, the year 2004 was not included because, during the monitoring phase, pit falls were installed in accordance with the agricultural year from September to July of the next year. Thus, pitfalls were also not installed over the entire year in 2004.

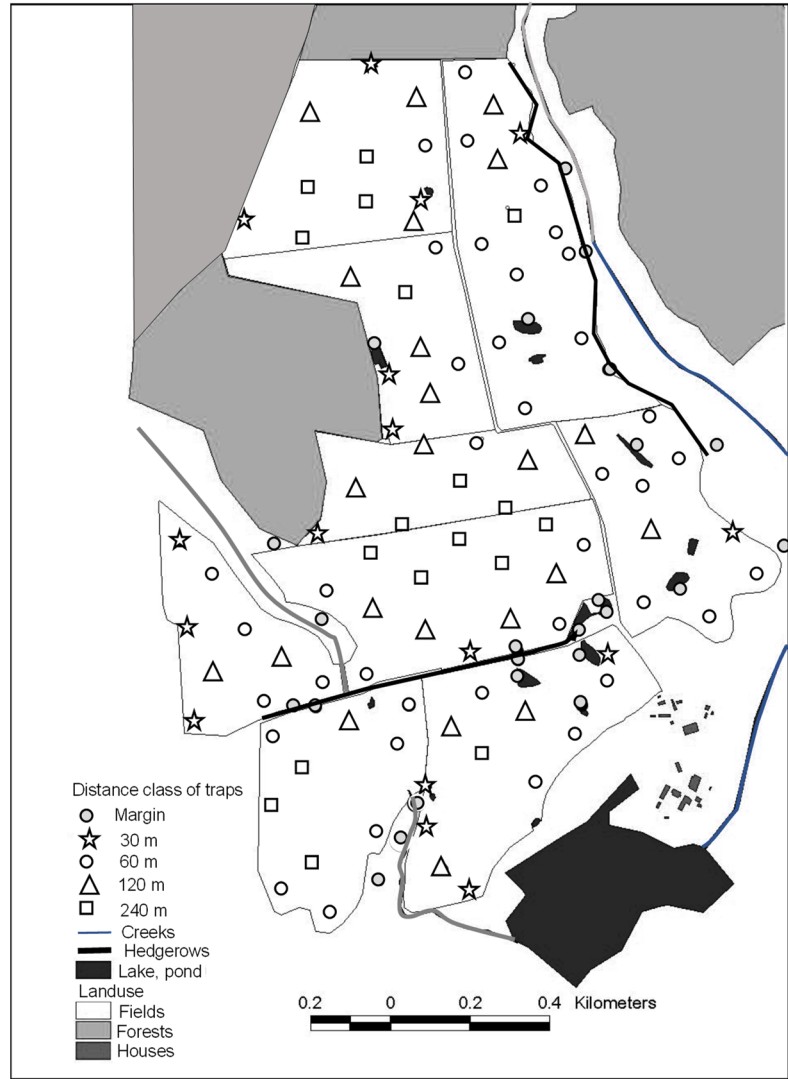

**Figure 1.** Map of the study fields with grid and distance classes of pitfall traps.

Statistical calculations were made using the program PAST 3 [13]. The seasonality of carabids was characterized using the weighted mean of yearly occurrence. The weighted mean (wM) was calculated as:

$$wM = \Sigma \ (n * month \ (1\text{-}12)/N); \qquad (1)$$

n = abundance in month (1–12), N = sum of total abundance.

The weighted mean was determined for groups of pitfall traps at a specific distance from the field margin. The following groups were discerned: 0 = traps outside the field in near-natural habitats, 30 = traps at >0–30 m distance from the field margin, 60 = traps at >30–60 m distance from the field margin, 120 = traps at >60–120 m distance from the field margin, and 240 = traps >120 m distance from the field margin. If a linear relationship was found between the wM and the distance from field margin, bivariate general linear model relations were calculated. According to the PAST handbook, this regression is robust to outliers. The algorithm employed was a "least trimmed squares" algorithm. Parametric error estimates are not available, but PAST gives bootstrapped confidence intervals for the slope and intercept. For logarithmic relationships, a logarithmic transformation of the data was performed. It was assumed that the wM differences between the distance groups indicate immigration or emigration. Therefore, the increment of the slope was

used to calculate the immigration or emigration speed of the species during seasonal migration processes.

According to [14], the foraging potential of a species was assumed to correspond with the activity-density determined by pitfall traps. For the analysis of foraging, only the data from 2015/16 and 2017/18 were selected to avoid the abundance changes caused by immigration or emigration processes. Foraging potential depends on the seasonal activity period and the area that the species covers during the activity period. The area covered by the species was estimated by counting the number of pitfall traps with records of the species. Thus, one pitfall trap in the field represents an area of 180 ha/96 pitfall traps = 1.9 ha. For the field margin traps, the corresponding value was 18 ha/27 = 0.7 ha. The effect of seasonality was calculated as the used monthly space (ST):

$$Points\ (\%) = number\ of\ pitfall\ traps\ with\ records\ 22 * 100/(27, 96) \tag{2}$$

$$Used\ space\ (ha) = (0.7, 1.9) * points\ (\%)/100 \tag{3}$$

$$AM = \Sigma\ (points\ in\ month\ (1\text{-}12)/(27, 96); \tag{4}$$

*AM*: part of space with activity of the total available space for a margin or field

$$ST = Used\ space * AM \tag{5}$$

*ST* is taken as an equivalent for the foraging effect and expresses the potential effect, but not the real foraging effect. The values were calculated separately for field traps and margin traps to evaluate the importance of organic fields for species living permanently in the arable field and those that migrate between a field and a margin, or between a margin and a field. Unweighted Euclidean pair group clustering was performed with the foraging effects for the margin and the field to differentiate species groups with similar foraging effects. Clusters were joined based on the average distance between all members in the two groups.

## 3. Results

### 3.1. Seasonal Occurrence and Migration Speed

For the present analysis on both fields and marginal habitats, 198,487 specimens and 36,303 specimens, respectively, were recorded. A total of 97 carabid species was recorded during the investigations from 2002 to 2018. Among them, 75 species were recorded from the arable field and 88 species from field margins. Most of these were only rarely found. As usual in ecological studies, several species were only found once, with a few records or in single years. These rare species were omitted from the present analysis because they have no influence on the foraging results due to their rarity. Overall, 56 frequent species were selected to analyze seasonality (Table 1). For six species, two yearly peaks of activity were clearly developed, usually with one peak higher than the second. The weighted monthly mean (wM) of activity over all species with only one peak was 6.1 ± 0.7. Thus, the end of May/beginning of June is the period with the highest number of active species. Six species were designated as typical summer species with a weighted monthly mean of activity in July. Eight species can be regarded as winter or early spring active species. Their weighted monthly means were either during winter months or, at the latest, in April. All species with two activity peaks belonged to this group, as the first peak was early in the year and the second peak in late autumn.

**Table 1.** Indicators of seasonality for selected species. W. mean: weighted mean of months with activity; for species with two distinct peaks of activity per year, both peaks are listed separately (bold type notes the major peak); Slope: slope for species with gradient w. mean in the margin—field gradient; s.d.: standard error for species without different change in w. mean in the distance gradient; 100 m (d): calculated speed of immigration with respect to emigration; *p*: probability error for distance (d); m: w. mean of earliest/latest activity peak; d group: distance group of earliest/latest activity peak.

| Species | W. Mean | Slope | 100 m | *p* | Beginning at | | Ending at | |
|---|---|---|---|---|---|---|---|---|
| | (Months *) | S.d. | (d) | | m | d Group | m | d Group |
| **no seasonal difference among distances** | | | | | | | | |
| *Bembidion obtusum* | **2.8**/11.6 | 0.18 | | | | | | |
| *Poecilus versicolor* | 5.4 | 0.05 | | | | | | |
| *Poecilus cupreus* | 5.5 | 0.03 | | | | | | |
| *Carabus auratus* | 5.5 | 0.07 | | | | | | |
| *Amara aenea* | 5.8 | 0.06 | | | | | | |
| *Loricera pilicornis* | 6.0 | 0.07 | | | | | | |
| *Bembidion lampros* | 6.1 | 0.15 | | | | | | |
| *Harpalus affinis* | 6.1 | 0.07 | | | | | | |
| *Agonum mülleri* | 6.1 | 0.09 | | | | | | |
| *Harpalus rubripes* | 6.2 | 0.39 | | | | | | |
| *Poecilus lepidus* | 6.3 | 0.22 | | | | | | |
| *Calosoma maderae* | 6.3 | 0.27 | | | | | | |
| *Amara lunicollis* | 6.3 | 0.19 | | | | | | |
| *Pterostichus melanarius* | 6.8 | 0.09 | | | | | | |
| *Pterostichus niger* | 7.3 | 0.06 | | | | | | |
| *Calathus cinctus* | 3.6/**10.5** | 0.11 | | | | | | |
| *Nebria salina* | 4.9/**10.7** | 0.16 | | | | | | |
| *Trechus quadristriatus* | 3.4/**10.8** | 0.19 | | | | | | |
| **seasonal occurrence increasing from center to margin** | | | | | | | | |
| *Acupalpus meridianus* | **5.0**/11.6 | −0.012 | 35.4 | 0.037 | 5.2 | 240 | 8.0 | 0 |
| *Clivina fossor* | 5.9 | −0.002 | 5.1 | 0.001 | 5.6 | 240 | 6.0 | 0 |
| *Harpalus rufipes* | 6.7 | −0.002 | 6.3 | 0.001 | 6.3 | 240 | 6.8 | 0 |
| *Harpalus signaticornis* | 6.7 | −0.003 | 9.9 | 0.050 | 6.0 | 240 | 6.8 | 0 |
| *Synuchus vivalis* | 7.1 | −0.004 | 11.7 | 0.001 | 6.5 | 240 | 7.5 | 0 |
| *Blemus discus* | 7.5 | −0.003 | 9.1 | 0.001 | 7.2 | 240 | 7.9 | 0 |
| *Calathus fuscipes* | 7.6 | −0.002 | 5.1 | 0.001 | 7.4 | 240 | 7.8 | 0 |
| **seasonal occurrence increasing from margin to center** | | | | | | | | |
| *Asaphidion flavipes* | 4.4 | 0.0082 | 24.6 | 0.030 | 4.1 | 0 | 6.0 | 240 |
| *Nebria brevicollis* | **4.9**/10.0 | 0.0051 | 15.3 | 0.001 | 4.2 | 0 | 5.5 | 240 |
| *Bembidion properans* | **5.4**/10.7 | 0.0017 | 5.4 | 0.003 | 5.1 | 0 | 5.5 | 240 |
| *Carabus nemoralis* | 5.4 | 0.0017 | 5.4 | 0.007 | 5.4 | 0 | 5.8 | 240 |
| *Agonum sexpunctatum* | 5.5 | 0.0029 | 8.7 | 0.048 | 5.2 | 0 | 5.9 | 240 |
| *Pterostichus anthracinus* | 5.6 | 0.0044 | 13.2 | 0.007 | 5.4 | 0 | 6.4 | 240 |
| *Carabus granulatus* | 5.7 | 0.0016 | 4.7 | 0.050 | 5.6 | 0 | 6.0 | 240 |
| *Limodromus assimilis* | 5.7 | 0.0012 | 3.5 | 0.050 | 5.6 | 0 | 5.8 | 240 |
| *Anchomenus dorsalis* | 5.9 | 0.002 | 2.0 | 0.006 | 5.8 | 0 | 6.0 | 240 |
| *Stomis pumicatus* | 6.0 | 0.0019 | 5.7 | 0.040 | 5.7 | 0 | 6.1 | 240 |
| *Abax parallelpipedus* | 6.0 | 0.0073 | 21.9 | 0.001 | 5.8 | 0 | 7.6 | 240 |
| *Amara similata* | 6.4 | 0.0002 | 2.0 | 0.001 | 6.3 | 0 | 6.4 | 240 |
| *Agonum viduum* | 6.4 | 0.0133 | 39.9 | 0.001 | 5.6 | 0 | 8.7 | 240 |
| *Amara plepebja* | 7.3 | 0.002 | 4.8 | n.s. | 7.1 | 0 | 7.5 | 240 |
| **logarithmic (ln) increase from margin to center** | | | | | | | | |
| *Bembidion tetracolum* | **4.4**/11.0 | 0.0986 | 13.6 | 0.011 | 4.0 | 0 | 4.6 | 240 |
| *Demetrias atricapillus* | 5.2 | 0.1896 | 6.3 | 0.001 | 4.6 | 0 | 5.6 | 240 |
| *Amara familiaris* | 5.8 | 0.1457 | 20.1 | 0.001 | 5.2 | 0 | 5.9 | 240 |

**Table 1.** *Cont.*

| Species | W. Mean | Slope | 100 m | *p* | Beginning at | | Ending at | |
|---|---|---|---|---|---|---|---|---|
| | (Months *) | S.d. | (d) | | m | d Group | m | d Group |
| *Trechoblemus micros* | 6.3 | 0.5331 | 73.6 | 0.009 | 4.4 | 0 | 7.3 | 240 |
| *Microlestes minutulus* | 6.4 | 0.1299 | 17.9 | 0.020 | 5.9 | 0 | 6.3 | 240 |
| seasonal occurrence increasing from margin but not reaching center | | | | | | | | |
| *Pterostichus strenuus* | **4.7**/10.8 | 0.0071 | 21.3 | 0.001 | 4.6 | 0 | 5.5 | 120 |
| *Bembidion lunulatum* | 4.7 | 0.0032 | 9.6 | 0.004 | 4.6 | 0 | 5.3 | 120 |
| *Acupalpus exiguus* | **5.3**/11.6 | 0.0081 | 29.2 | 0.044 | 5.2 | 0 | 6.1 | 120 |
| *Pterostichus oblongipunctatus* | 5.4 | 0.066 | 16.5 | 0.050 | 5.1 | 0 | 5.7 | 120 |
| *Pterostichus diligens* | 5.4 | 0.0073 | 21.9 | 0.025 | 5.0 | 0 | 5.8 | 120 |
| *Bembidion guttula* | 5.6 | 0.0129 | 38.7 | 0.001 | 5.2 | 0 | 6.2 | 120 |
| *Notiophilus biguttatus* | 6.1 | 0.0028 | 8.4 | 0.001 | 6.0 | 0 | 6.7 | 120 |
| *Amara communis* | 6.6 | 0.012 | 37.5 | 0.003 | 5.8 | 0 | 7.3 | 120 |
| *Dyschirius globosus* | 6.6 | 0.0051 | 15.3 | 0.001 | 6.1 | 0 | 6.7 | 120 |
| *Pterostichus nigrita* | 6.8 | 0.0067 | 45.9 | 0.011 | 6.0 | 0 | 7.8 | 120 |
| *Pterostichus vernalis* | 7.3 | 0.0068 | 20.4 | 0.050 | 7.2 | 0 | 7.9 | 120 |
| *Agonum fuliginosum* | **5.3**/10.7 | 0.0177 | 53.1 | 0.047 | 5.1 | 0 | 6.2 | 60 |

* For species with two clearly separate seasonal peaks per year the main peak is highlighted by bold digits and used for further calculations.

Among the species analyzed, 18 species showed no difference along the distance gradient from the margin to the field. This involves 32% of the species examined. It can be assumed that these species either overwinter in the arable field or the migration process is so fast that the monthly resolution of the sampling periods is too low. For another seven species (12%), the activity peak increased temporarily from the field center to the margin. In contrast to the first species group, these species seemed to overwinter in the field center and likely not in the marginal habitats. For most species of the group, the migration speed was fast, ranging between approximately 5 and 10 days for 100 m (with one exception). However, if these species overwinter in the entire field area, the distance to the margin must be low. A large group of 28 species (50%) immigrated from the margins into the arable field. Many species potentially reached the field center within one year. They were rather fast migrators, and only needed $11.7 \pm 11.2$ days for 100 m of immigration. The mean starting month for their migration was about mid-May. Few species showed a logarithmic change of seasonality. Their immigration speed was slower, with $26.3 \pm 27.0$ for 100 m of immigration, but the starting time became earlier at end of April/beginning of May. The last group of species never reached the center of the arable fields. They were also slow migrators, with $26.5 \pm 14.4$ days, but their starting time was in the second half of May. They could potentially reach the 120 m distance group. Among these species, one species, *Agonum fuliginosum*, only reached the 60 m distance group. It was the species with the slowest speed among all the analyzed species, at 53 days for 100 m distance.

### 3.2. Time and Space Related Foraging Potentials

The foraging effect of carabids on the arable fields and their adjacent margins depends on the time of activity during the yearly season and the space, which they cover during their activity period. In Figure 2, seasonal activity is illustrated for four species in the two years studied. *Carabus auratus* covers nearly the entire arable field (178 ha of 180 ha), but has only a short time of activity (Figure 2). In contrast, *Bembidion tetracolum* covers nearly the same space on the arable fields (180 out of 180), but has a much longer activity period. The same applies to species of the field margins such as *Limodromus assimilis* with a short activity period, and *Carabus nemoralis* with a long activity period. In Table 2, the estimated results of the space effect covered by a species during its activity period are listed.

**Table 2.** Values to estimate the effect of spatial and temporal use by the selected species. AM: part of space with activity of the total available space for margin or field; ST: equivalent for temporal and spatial use; potential use was calculated for an equally high use of 100% over the whole year.

| Species | Margin | | | | Field | | | | Potential Use | |
|---|---|---|---|---|---|---|---|---|---|---|
| | Points (%) | Space (ha) | AM | ST | Points (%) | Space (ha) | AM | ST | Margin (%) | Field (%) |
| Group 1 with equal use of margin and field | | | | | | | | | | |
| *Trechus quadristriatus* | 100 | 18 | 9.4 | 14.1 | 100 | 180 | 10.1 | 151.1 | 52 | 84 |
| *Bembidion tetracolum* | 100 | 18 | 10.0 | 15.1 | 100 | 180 | 9.3 | 139.1 | 56 | 77 |
| *Pterostichus melanarius* | 100 | 18 | 7.8 | 11.7 | 100 | 180 | 8.3 | 124.7 | 43 | 69 |
| *Anchomenus dorsalis* | 100 | 18 | 8.1 | 12.2 | 100 | 180 | 7.0 | 105.3 | 45 | 59 |
| Group 2 with equal use of margin and field and lower foraging effect | | | | | | | | | | |
| *Agonum mülleri* | 100 | 18 | 5.4 | 8.2 | 100 | 180 | 7.4 | 111.7 | 30 | 62 |
| *Bembidion lampros* | 96 | 17 | 5.5 | 7.8 | 100 | 180 | 6.6 | 99.7 | 29 | 55 |
| *Nebria brevicollis* | 100 | 18 | 6.6 | 9.9 | 100 | 180 | 6.0 | 89.5 | 37 | 50 |
| *Carabus granulatus* | 100 | 18 | 6.1 | 9.1 | 100 | 180 | 5.5 | 82.5 | 34 | 46 |
| *Nebria salina* | 93 | 17 | 3.8 | 5.4 | 100 | 180 | 6.5 | 98.1 | 20 | 55 |
| *Poecilus cupreus* | 100 | 18 | 4.4 | 6.6 | 100 | 180 | 6.4 | 96.1 | 24 | 53 |
| *Harpalus affinis* | 100 | 18 | 3.1 | 4.7 | 100 | 180 | 5.9 | 87.8 | 17 | 49 |
| Group 3 with small distribution and small foraging effect | | | | | | | | | | |
| *Clivina fossor* | 93 | 17 | 3.5 | 5.0 | 99 | 178 | 4.2 | 62.6 | 18 | 35 |
| *Poecilus versicolor* | 96 | 17 | 4.0 | 5.7 | 100 | 180 | 4.1 | 61.7 | 21 | 34 |
| *Loricera pilicornis* | 100 | 18 | 3.0 | 4.6 | 100 | 180 | 4.1 | 60.9 | 17 | 34 |
| *Amara similata* | 100 | 18 | 3.4 | 5.2 | 100 | 180 | 3.8 | 57.7 | 19 | 32 |
| *Bembidion properans* | 59 | 11 | 2.0 | 1.8 | 95 | 171 | 3.5 | 49.3 | 7 | 27 |
| *Amara familiaris* | 85 | 15 | 2.0 | 2.5 | 100 | 180 | 3.1 | 47.2 | 9 | 26 |
| *Notiophilus biguttatus* | 78 | 14 | 4.1 | 4.8 | 95 | 171 | 3.3 | 47.1 | 18 | 26 |
| *Harpalus rufipes* | 89 | 16 | 2.7 | 3.7 | 99 | 178 | 3.1 | 46.1 | 14 | 26 |
| *Calathus fuscipes* | 74 | 13 | 1.9 | 2.1 | 92 | 165 | 3.1 | 43.3 | 8 | 24 |
| *Carabus auratus* | 81 | 15 | 1.6 | 1.9 | 95 | 171 | 3.0 | 43.1 | 7 | 24 |
| *Amara aenea* | 52 | 9 | 0.9 | 0.7 | 93 | 167 | 2.5 | 35.3 | 3 | 20 |
| *Calathus cinctus* | 63 | 11 | 1.2 | 1.1 | 80 | 144 | 2.4 | 29.1 | 4 | 16 |
| *Harpalus signaticornis* | 67 | 12 | 1.7 | 1.7 | 91 | 163 | 2.1 | 28.6 | 6 | 16 |
| *Demetrias atricapillus* | 78 | 14 | 2.1 | 2.5 | 91 | 163 | 1.8 | 24.5 | 9 | 14 |
| *Pterostichus niger* | 100 | 18 | 4.3 | 6.5 | 100 | 180 | 2.6 | 38.8 | 24 | 22 |
| *Limodromus assimilis* | 85 | 15 | 3.9 | 5.0 | 78 | 141 | 1.8 | 21.1 | 18 | 12 |
| Group 4 with high foraging effects in the margin | | | | | | | | | | |
| *Pterostichus strenuous* | 100 | 18 | 6.6 | 9.9 | 54 | 98 | 1.0 | 8.5 | 37 | 5 |
| *Carabus nemoralis* | 96 | 17 | 6.7 | 9.5 | 95 | 171 | 3.1 | 44.4 | 35 | 25 |
| Group 5 with small foraging effects margin and field | | | | | | | | | | |
| *Amara plepebja* | 81 | 15 | 1.7 | 2.1 | 74 | 133 | 1.4 | 15.5 | 8 | 9 |
| *Bembidion obtusum* | 59 | 11 | 1.3 | 1.1 | 65 | 116 | 1.7 | 16.5 | 4 | 9 |
| *Agonum sexpunctatum* | 56 | 10 | 1.1 | 0.9 | 68 | 122 | 1.1 | 11.1 | 3 | 6 |
| *Acupalpus meridianus* | 33 | 6 | 0.5 | 0.2 | 58 | 105 | 1.3 | 11.0 | 1 | 6 |
| *Calosoma maderae* | 33 | 6 | 0.4 | 0.2 | 61 | 111 | 1.2 | 10.7 | 1 | 6 |
| *Microlestes minutulus* | 26 | 5 | 0.4 | 0.1 | 47 | 84 | 0.7 | 4.8 | 1 | 3 |
| *Blemus discus* | 19 | 3 | 0.2 | 0.1 | 32 | 58 | 0.5 | 2.6 | 0 | 1 |
| *Poecilus lepidus* | 4 | 1 | 0.0 | 0.0 | 29 | 53 | 0.5 | 2.1 | 0 | 1 |
| *Bembidion guttula* | 93 | 17 | 4.4 | 6.2 | 45 | 81 | 0.6 | 4.3 | 23 | 2 |
| *Pterostichus nigrita* | 89 | 16 | 4.3 | 5.7 | 44 | 79 | 0.8 | 5.3 | 21 | 3 |
| *Amara communis* | 85 | 15 | 1.7 | 2.2 | 15 | 65.6 | 0.5 | 2.6 | 8 | 1 |
| *Pterostichus anthracinus* | 78 | 14 | 4.1 | 4.8 | 30 | 54 | 0.6 | 2.7 | 18 | 1 |
| *Agonum viduum* | 100 | 18 | 2.6 | 3.9 | 22 | 39 | 0.3 | 0.9 | 15 | 0 |
| *Synuchus vivalis* | 56 | 10 | 0.9 | 0.7 | 10 | 96 | 0.8 | 6.2 | 3 | 3 |
| *Pterostichus vernalis* | 89 | 16 | 2.7 | 3.6 | 60 | 109 | 1.0 | 9.4 | 13 | 5 |
| *Pterosticus diligens* | 81 | 15 | 2.3 | 2.9 | 9 | 17 | 0.1 | 0.1 | 11 | 0 |

**Table 2.** *Cont.*

| Species | Margin | | | | Field | | | | Potential Use | |
|---|---|---|---|---|---|---|---|---|---|---|
| | Points (%) | Space (ha) | AM | ST | Points (%) | Space (ha) | AM | ST | Margin (%) | Field (%) |
| *Acupalpus exiguus* | 78 | 14 | 2.3 | 2.7 | 29 | 53 | 0.4 | 2.0 | 10 | 1 |
| *Abax parallelpipedus* | 67 | 12 | 2.2 | 2.2 | 29 | 53 | 0.4 | 1.5 | 8 | 1 |
| *Dyschirius globosus* | 70 | 13 | 2.1 | 2.2 | 11 | 21 | 0.1 | 0.2 | 8 | 0 |
| *Agonum fuliginosum* | 63 | 11 | 1.6 | 1.5 | 5 | 9 | 0.1 | 0.1 | 6 | 0 |
| *Amara lunicollis* | 67 | 12 | 1.4 | 1.4 | 21 | 38 | 0.3 | 0.8 | 5 | 0 |
| *Stomis pumicatus* | 63 | 11 | 1.1 | 1.0 | 46 | 83 | 0.7 | 4.9 | 4 | 3 |
| *Bembidion lunulatum* | 41 | 7 | 1.2 | 0.7 | 15 | 26 | 0.4 | 0.8 | 3 | 0 |
| *Harpalus rubripes* | 56 | 10 | 0.9 | 0.7 | 34 | 62 | 0.4 | 2.2 | 3 | 1 |
| *Asaphidion flavipes* | 37 | 7 | 0.6 | 0.3 | 15 | 26 | 0.1 | 0.3 | 1 | 0 |
| *Pt. oblongopunctatus* | 37 | 7 | 0.5 | 0.3 | 7 | 13 | 0.1 | 0.1 | 1 | 0 |
| *Trechoblemus micros* | 37 | 7 | 0.4 | 0.2 | 32 | 58 | 0.5 | 2.3 | 1 | 1 |

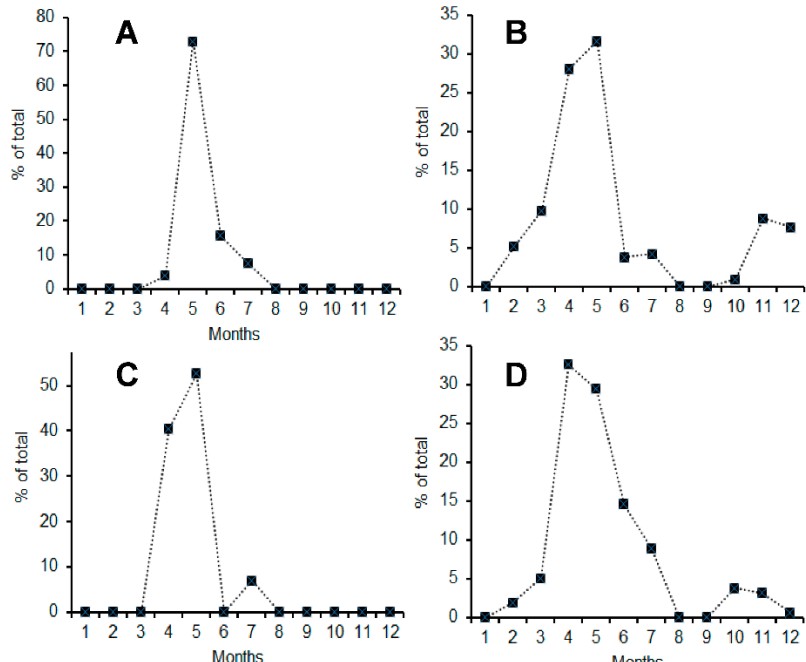

**Figure 2.** Seasonal activity changes for the two years, 2015/16 and 2017/18, with a short and long activity period of (**A**) *Carabus auratus* and (**B**) *Bembidion tetracolum* as typical species of the arable field and (**C**) *Limodromus assimilis* and (**D**) *Carabus nemoralis* of field margins with immigration into the field.

Regarding the potential foraging use of arable fields and margins, a large variety of foraging strategies can be found. There are species with a wide distribution and long seasonal activity, those with a wide distribution and short seasonal activity, and species with a small distribution and short seasonal activity. All transitional types appear. Nevertheless, five groups can be distinguished based on pair group clustering using the Euclidean similarity distance for the values of the potential use of the margin and the field (Figure 3).

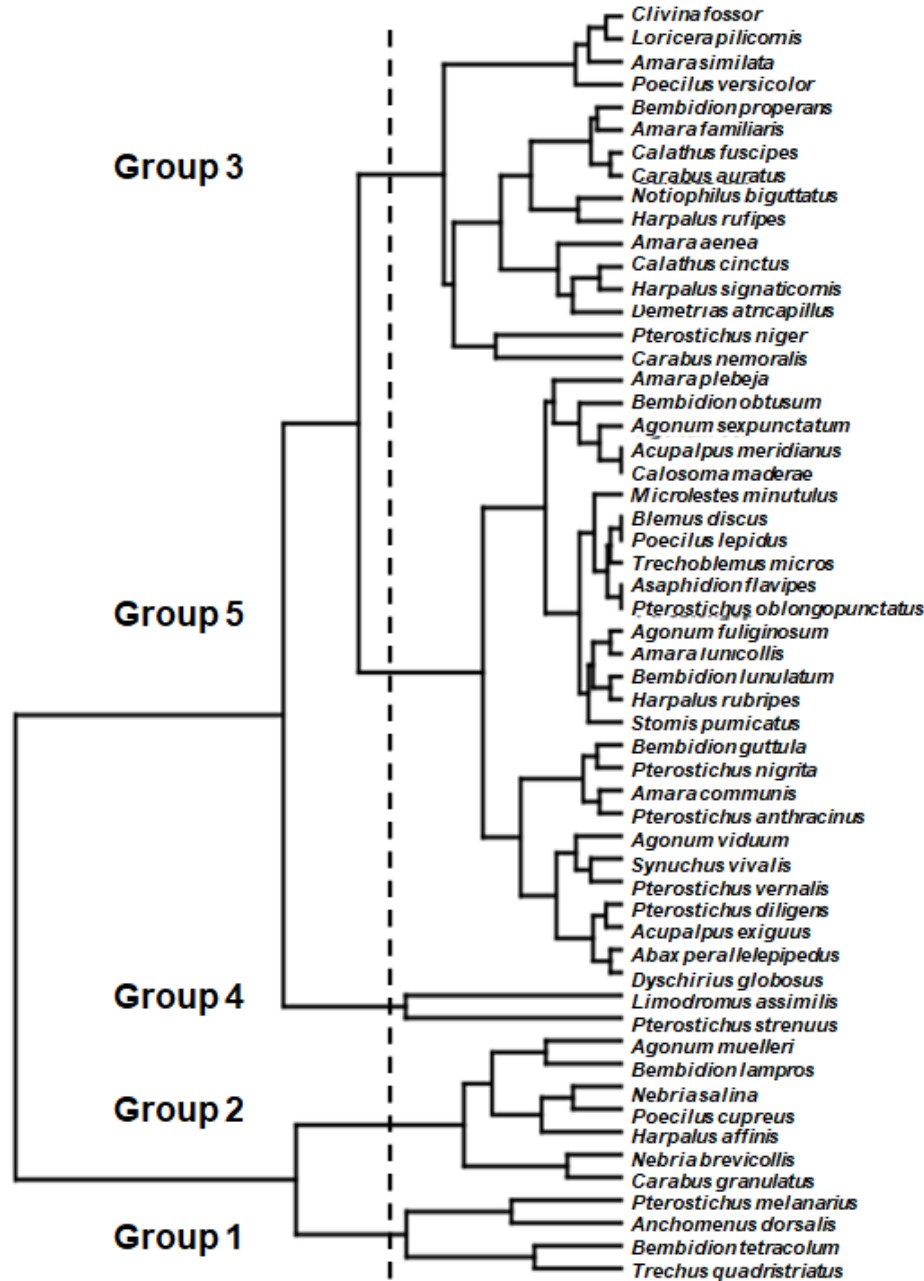

**Figure 3.** Cluster of species using values for potential forage use (pair-group clustering; Euclidean similarity distance) and separated groups.

The four species of the first group can use both arable fields and margins and, additionally, have a long seasonal activity period. Their foraging potentials range between 49% ± 6% and 72% ± 11% of the margins and the arable fields, respectively. Nearly all species of this group can be categorized as typical species of arable fields. Thus, their potential use of the arable fields is slightly higher than their use of the margin. The next group of seven species can also use nearly the whole study area, but their seasonal occurrence is much shorter. Therefore, their foraging potential is distinctly reduced to an average of 27% ± 7% of the margin and 53% ± 5% of the arable fields. Among these species, typical field species are found, such as *Harpalus affinis*, as well as species of the margins with a high dispersion potential, such as *Carabus granulatus*. A third group of sixteen species combines species that are characterized by a small distribution and a short seasonal occurrence in both margins and arable fields. Thus, their foraging effect is small, ranging between 13% ± 7% of the marginal area and 24% ± 7% of the arable field area. The next group consists of

only two species with major foraging potentials in the margins, but also still notable values in the fields. The range lies between 36% ± 1% for the margin and 15% ± 14% for the arable fields. The two species of this group are typical species of the field margin, but with a high migration speed. The following group of 27 species have only a low foraging potential in both margins and fields. Their activity period and their spatial distribution are extremely short. Here, typical species of the field margins and typical species of the arable fields are combined. In total, their mean foraging potential ranges between only 7% ± 7% for the margins and 2% ± 3% for the arable fields. The higher values for the marginal habitats indicate that the majority of the species in this group occur in the field margins. *Poecilus lepidus* alone can be categorized as a typical arable field species because its maximum spatial use of the margins (points%) is only 4, whereas the respective value of the arable field is 53. If the foraging effect of all carabids is set to 100% for both margins and fields, field and margin species (type one and two, Table 1) account for 63% and 37%, respectively. Separating these effects only for the field (setting foraging in the field to 100%), field species make up 65% of the total foraging, whereas margin species account for 35%.

## 4. Discussion

The changing composition of species in succession after the conversion from intensive to organic farming is a complex process, which is underlain by various factors [15]. It depends on the changes in the fields [7], the structure of the surrounding landscape [16], and the directly adjacent habitats [17]. To understand this process, long-term dispersion and seasonal migrations have to be distinguished. Long-term dispersions of carabids into the fields are affected by the population size in the surrounding landscape and the dispersion ability of the species [18]. For the studied fields, the composition of species according to their relation to field and marginal habitats has already been studied in detail for carabids [12]. Six types of species were separated: (1) formerly extremely abundant species with decreasing records; (2) silvicolous species with a decreasing abundance caused by the sunnier soil surface; (3) species of open habitats with an increasing abundance separated into fast immigrating species during the few years after conversion, intermediate species with immigration between 5 and 10 years after conversion, and later species immigrating after 10 years or even later; (4) typical species of arable fields but never occurring under intensive farming practice; (5) species of wet habitats that mainly invade in years with a high rainfall; (6) species that only occur accidently in arable fields.

Immigrating species of type three with intermediate or long intervals after conversion immigrated by pioneer specimens. This was, for example, observed for *Nebria salina* after a few years, *Poecilus lepidus* after 10 years, and *Zabrus tenebrioides* after 15 years [12]. These pioneer specimens are often flying specimens in a population of wingless individuals [19,20]. This type of dispersion must be separated from the dispersion of nearer located populations of the margins or nearby habitats. For these species, a normal random walk or directed movement into the field can be assumed [21]. Whereas the establishment of populations by pioneer species happens in species with arable field preference which remain for a long time, the species using the field only seasonally come from adjacent habitats of wet sites, grassland, and woody or shrubby sites. They use the fields for additional foraging.

In the present study on seasonality, species of the first five types were analyzed. Typical agrarian species showed no spatial differences in their seasonal occurrence. Among them were species of various types, e.g., *Pterostichus melanarius* (type one), *Poecilus lepidus* (type three with late immigration), or *Carabus auratus* (type four). Many of these species were able to equally use the field area and the adjacent margins. Others seemed to overwinter only in the field area, and could use the marginal habitats after a short time of movement to the margins. According to the results of the spatial differences in seasonal occurrence, the invasion process of species that seasonally invade into the field area depends on the time of seasonal activity and the dispersion speed. Species with fast-speed potentials were able to use the field even if their mean occurrence lay in the second half of May. The species with low-speed potentials needed an earlier occurrence, at least the end of April to mid-May.

Species with a low-speed ability with a mean occurrence in late May or June were not able to use the entire field area. Their maximum invasion from the margins ended at 120 m, before harvesting and other agricultural work started in August/September.

The immigration speed is important to estimate the foraging effects of species invading from the marginal habitats. In particular, in organic farming, the foraging of beneficial insects on pest insects is of great interest, because pesticides cannot be applied. Therefore, predation on aphids or leaf-mining flies or midges, e.g., by *Anchomenus dorsalis* [22], is a useful natural pest control. *Amara aenea* that seem to overwinter in the field is a beneficial species in pest control. They develop faster and grow larger if the insect diet is combined with seed food [23]. Many *Amara* species that invade from the field margin and were formerly known as seed feeders, such as *Amara similata* [24], were recently found to be omnivorous [25]. Large predators, such as *Carabus granulatus*, *Carabus auratus*, and *Pterostichus melanarius*, mainly feed on earthworms and slugs [26,27]. According to [14], the activity of *Carabus granulatus* closely corresponds with its foraging effect. Regarding the results of the studied region, the carabids from the margins made up about 35% of the total foraging effect of carabids in the organic field.

The effects of single species vary greatly and depend on their distribution and seasonal activity. For example, *Abax parallelpipedus* has a high preference for forest habitats, but can also use hedges as dispersing corridors [28]. From there, it also invades into adjacent fields during high activity periods, with high progress into cornfields and low progress into carrot fields [29]. According to [30], the daily consumption of the species amounts to 7–10% of its weight of 18 to 24 mg. Changes succeed during long-term succession, as shown by several species which invaded after the conversion from intensive to organic farming; for example, *Carabus auratus* was not present on the field under intensive agriculture. Now the species accounts for 2% of the total foraging effect. *Poecilus lepidus* immigrated after more than 10 years [12], and the dispersion process is still not complete after nearly 20 years. Thus, the 0.1% of foraging effects will certainly increase in the future. Nevertheless, typical field species, such as *Pterostichus melanarius* and *Trechus quadristriatus*, have strong effects amounting to 6% to 7% of total foraging. In addition to the named factors that regulate foraging, size and food resources play an important role in estimating the effect of carabids. According to [26], carabids consume between 0.75 and 1.00- and up to 3.4-times their own weight daily. As many large species dispersed into the field or within the field after conversion, the benefit for organic farming of invading species or dispersing species from the margin to the field must be interpreted as a notable effect for pest control in organic farming.

**Funding:** This research was funded privately by Günther Fielmann.

**Institutional Review Board Statement:** Ethical review and approval were waived for this study due to missing reasons by the record of data.

**Informed Consent Statement:** Not applicable.

**Data Availability Statement:** Data are available in the database of Institute for Ecosystem Research.

**Acknowledgments:** My sincere thanks are extended to Günther Fielmann, who enabled and financially supported the research on his fields of Ritzerau Manor. In particular, his high interest in the study results was stimulating. Thanks are also due to the manager of the farm, Martin Natmeßnig, for his steady support of the study in the field and his amicable help. I am also grateful to many students, who helped during nearly 20 years of monitoring research and cannot all be individually named here. I am deeply indebted to Lars Schröter, who performed the investigations from 2001 to 2010, and to Jean Heitmann, who carried out the fieldwork from 2010 to 2018.

**Conflicts of Interest:** The author declares no conflict of interest.

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
