# Peer review of "Seasonality of Carabid Beetles on an Organic Agricultural Field and Its Effect on Foraging Use"

_agriculture, doi:10.3390/agriculture12050596_

Round 1
Reviewer 1 Report
General comments
The manuscript presents a long-term study on carabid beetle in an organic field and in marginal near-natural habitats. The major aims of the study are to analyse the speed of seasonal immigration into field and what effect the seasonal activity of species has on foraging use. This is an interesting and important topic, which falls within the general scope of the journal. The title of the manuscript well reflects its contents, the abstract is informative.
The introduction might be somewhat enlarged; the research objectives are well formulated. However, particularly material and methods and results have significant flaws and should be improved (see detailed comments). To be honest, the manuscript makes the impression that it was written under a strong time pressure.
The list of references includes 31 positions, which seem to cover well the knowledge about the research topic. However, not all of them are mentioned in the running text of the manuscript (see detailed comments). The paper is supplemented by two tables and two figures, all of them important for the manuscript.
Detailed comments
1) Introduction, lines 24-37: The introduction starts with a “simple summary”. I assume this is a mistake with respect to text formatting – papers already published in “Agriculture” do not have such a “simple summary” at the beginning of the introduction.
2) Introduction line 44: Günther Fielmann is well known in Germany, but probably not in many others countries. I propose to provide with a short explanation who is Günther Fielmann.
3) Introduction, lines 46-48: A bit more information about the research area might be useful in order to enable the reader to better understand the research questions. Lines 58-65 present some basic information. Maybe, it might be of sense to shift some of this information to the introduction.
4) Introduction, line 55: Should be “…seasonal immigration into the field…”.
5) Methods and sites, line 71: The author mentions “Within the study period…”. Please specify the years.
6) Methods and sites, lines 79-82. The trap scheme is not clear. How many lines and rows had the grid? How were the traps in the near-natural habitat arranged? In my opinion, a map with the location of the traps would be very useful.
7) Methods and sites, line 96: Which version of PAST was used? Please specify. The author mentions for the software PAST the reference number “[12]”; however, the respective reference has nothing in common with this software. In the list of references, the adequate reference has the number [17]. Moreover, the highest number in the text of the manuscript is [26], but the reference list has 31 positions. Thus, 5 references are not mentioned in the text. According to that the difference between [12] and [17] is also 5, implying a relation between these two mistakes.
8) Methods and sites, lines 116-117: Without a scheme showing the shape of the sites and the location of the traps it is not clear, why the area represented by a trap can be calculated by simply diving the whole area of the site through the numbers of traps. However, the area covered by the individual species is a key aspect of the manuscript. Therefore, the author has to show clearly that the above mentioned method is valid.
9) Methods and sites, line 129: The dot is missing at the end of the sentence.
10) Results, line 132: Why does the author refer to the period 2001-2018? In the methods he states that “for the present study, the data from 2002 to 2018 was used” (line 91).
11) Results, lines 132-133: In my opinion it will be also interesting for the readers how many individuals were collected.
12) Results, lines 134-136: Based on which criteria the 56 species in the table were selected? Please specify. This is a quite important issue because many aspects of results and discussion focus on these 56 species.
13) Results, headline to table 2, line 182: It should be also mentioned that the table shows only selected species.
14) Results, figure 1: The label of the y-axis is “% of total”. Total of what? Please specify.
15) Discussion, line 279: “(Húrka and Jarosík, 2003)” has to be replaced by “[18]”.
Conclusion
The authors present a paper, which is interesting and scientifically important. However, There are several flaws, particularly with respect to the methods description, the presentation of the results and the assignment of references. Corrections and improvements of the manuscript are needed in order to fully understand the significance of the results. Therefore, I recommend a major revision of the manuscript.
Author Response
Dear Reviewer 1
- This was a misinterpretation. The ms was originally written for “Insects” and they want to have a “Simple Summery” and an “Abstract”. I cancelled one of it and entitle it “Abstract”. I hope this is now within the scope of “Agriculture”.
- Corrected considering the remarks of reviewer 2 in: … provoked the owner of the farm ..
- The following additional information is added: “The results showed that after 10 years biodiversity has increased mainly from the margins [16]. The invading species originated from open habitats, whereas silvicolous species retreated [5, 14]. Additionally, earthworm fauna changed with increasing density of anecic species indicating also changes in the waterblance of soils [15].”
- corrected
- corrected
- A map is added with classification of trap distances and locations
- I apologise for the mistake in the references. This is due to the automatic transformation to reference numbers.
- The map is added showing that traps cover the field area in more or less equal distances. Naturally, the individual area of traps may slightly differ. But it is also true that GPS data have inaccuracy of at least 5 meters and several more errors may influence the exact area around single traps. However, the area around traps estimated to be 1.9 ha on the fields may differ not more than 0.1 to 0.2 ha which can be neglected in the foraging estimation. Moreover, the present study can only separate the relative differences between field species and marginal species. Naturally, real foraging depends on multiple factors that cannot all be considered, e.g. abundance in the different years, local climate, individual hunger of animals etc. Here only a rough estimation was tried which reflects certainly only a rough image of the real feeding processes on the field. There are two approaches to this problem: either a very detailed approach, which can only refer to very few species such as done for Carabus granulatus in [8] or Agonum dorsale in [19] or a more less exact approach as in the present study but under consideration of the whole carabid assemblage.
- Corrected
- The year 2001 was omitted because data were not available for the whole year. This is the next sentence after your cited text: for the present study, the data from 2002 to 2018 was used.
- Number of specimens added: For the present analysis on both fields and marginal habitats 198,487 specimens and 36,303 specimens, respectively, were recorded.
- I added: As usual in ecological studies, several species were only found once, with a few records or in single years. These rare species were omitted from the present analysis because they have no influence on the foraging results due to their rarity.
- Corrected
Numbering of reference was corrected
Reviewer 2 Report
General comments
The manuscript (ms) submitted a full length article to the journal. The study seems well designed agroecological study on carabids and their potential spillover between organic croplands and their corresponding crop margins. The study identified differently dispersed set of species in the study area, based on a long-term rigorous sampling schemes. Although the ms is well-organized, I have several issue identified in terms of concept, design, and writing. The language of the manuscript a bit clumsy some occasions, the terms and phrases used which are not line line with the expectation in agroecology papers. I’m not fully convinced that the study is based on a spatially replicated sampling scheme, neither the methods, nor the results do not help to find an evidence for unfolding this issue. In addition, the applied statistical methods seem a bit overcomplicated, those are correct in terms of implementation but it is difficult for the reader to decode the information, due to the fact that the majority of the results are in Tables.
Specific comments
Line 22 - “Seasonal immigration” - this phenomenon called spillover, I admit the fact that the author used a long term dataset, but the spillover simply fits better to describe the situation.
Line 44 – In my opinion should avoid and direct advertisement of a private land owner and his effort in organic agriculture. The ms cites already published papers from this organic farm, thus those would be better to cite and focus of the organic management and not for who made it.
Line 79-82 & 101-106 – Is there any true spatial replicates include in the pitfall trap sampling?
Line 99 - This weighted means looks like a relative abundance formula. The sampling design may suggest a rigours sampling scheme with repeated measures (sampling same location the same period of the consecutive years), but the year wight function in the formula may indicate a varying sampling intensity between years. This issue should be clearly describe the text above.
Line 106 - Please describe “general linear model relations” in detail. Please focus on the response and predictor variables, including error terms and standardisation.
Line 115-116 - Please add some citation for spatial constrain of pitfall traps.
Line 127-128 - Is this a hierarchical clustering? If so please add all the parameters including similarity functions and functional algorithm for cluster drawing.
Line 144-147 - Please add all the abbreviation for the table legend. What test was the basis of the p-values?
Line 189-190, Fig 1 - What kind of data were used for this graph? One particular year or mean values between 2001-2018? If the later one, where is the dispersion parameters(SD or SEM) on the graphs?
Author Response
Dear Reviewer 2
- I do not agree with the reviewer. The observed effect is not a spillover. It is a seasonal phenomenon that occurs in every year. It is, however, true that in some species the immigration looks like a spillover, in particular, if immigrations happen with higher abundance only in wet years for species with wet preference. But mostly it is a yearly repeated immigration and caused by the normal dispersion behaviour of the species, see also my publication [4].
- This is in agreement with reviewer 1 and corrected.
- But what is a replicate in the pitfall trap method? It is impossible to place two pitfall traps at the same place. Replicates in the present investigation is represented by the grid arrangement of the traps. The main idea behind the investigation was to study the spatial and temporal changes of carabids and this is only possible using the selected grid arrangement.
- Weighted mean is totally different from relative abundance. Using relative abundance, has the aim to find out differences in abundance between species. Weighted mean, here, has the aim to find out differences in seasonal occurrence. Two species with same seasonal occurrence may totally differ in their relative abundance.
- I added the information found in PAST handbook to the selected modell: According to PAST handbook the regression is robust to outliers. The algorithm is “Least Trimmed Squares” Parametric error estimates are not available, but PAST gives bootstrapped confidence intervals on slope and intercept (beware.
- I do not understand what the reviewer wants to hear. It is a simple calculation performed with Excel and need no citation. I hope the process becomes clearer by the added map with distance classification of traps.
- It was an Unweighted Euclidean pair group clustering … and “Clusters are joined based on the average distance between all members in the two groups.”
- Explanation for added
I corrected it. This addition was also made in “methods” because it is valid for all calculations of the foraging part. The graphs too refer to the last study years 2015/16 and 2017/18.
Reviewer 3 Report
This is a well written paper describing an ecological study on the seasonal occurrence of carabid species on an organically cultivated field. Moreover, the author has investigated the migration speed of the different ground beetle species invading from the marginal habitats. This work is a follow-up of previous publications of the author on this research topic. The author has done a tremendous amount of work over an 18 years-long study period. During this period, a huge number of specimens has been collected and identified. The study seems to be well-executed and analyzed.
My general impression of this study and its presentation in the form of a manuscript is positive. I only got confused by the numbering of the references. For instance, in the introduction section, just after the first three publications [1-3], the author “jumps” to references 11 and 12 [11,12]. I suggest the author goes through the whole manuscript, renumbering the references, as well as checking their relevance.
Line 40: Replace “waterblance” with “Water balance”.
Author Response
Reviewer 3 (round 1)
My general impression of this study and its presentation in the form of a manuscript is positive. I only got confused by the numbering of the references. For instance, in the introduction section, just after the first three publications [1-3], the author “jumps” to references 11 and 12 [11,12]. I suggest the author goes through the whole manuscript, renumbering the references, as well as checking their relevance.
Numbering of references was corrected according to your proposal
Line 40: Replace “waterblance” with “Water balance”.
corrected
Round 2
Reviewer 1 Report
The author addressed my comments on the manuscript very well. Therefore I can recommend the manuscript for publication.
The only aspect which still should be controlled is the numeration of the references. In lines 35-36 is the numeration “[16, 35 14, 5, 15, 31]”; next is the number [17] (line 86). I think that from formal viewpoint some corrections are necessary.
Author Response
Reviewer 1 (round 2)
The only aspect which still should be controlled is the numeration of the references. In lines 35-36 is the numeration “[16, 35 14, 5, 15, 31]”; next is the number [17] (line 86). I think that from formal viewpoint some corrections are necessary.
Numbering of references was corrected according to your proposal
Reviewer 2 Report
I have received the 1st revision from the suggested manuscript. I’m regret to report that the quality of the manuscript do not meet the criteria at the Journal Agriculture. My suggestion is based on the following reason below:
1. Author cannot provide eligible documentation of the changes in the text and figures during the revision. The replies are rather short and general there is no clear professional/scientific reasoning therein. Some of the replies are under the level of the expected quality of the language that supposed to be used in scientific communication. This issues supposed to be controlled by the author firmly. The author did not thoroughly revised the manuscript, some major issues have replied, some of them are ignored without any reason.
2. The author does not know the concept of true spatial replicates in ecological studies to avoid pseudoreplication (Hurlbert, Stuart H. “Pseudoreplication and the Design of Ecological Field Experiments.” Ecological Monographs 54, no. 2 (1984): 187–211. https://doi.org/10.2307/1942661.). In addition, the author was not able to provide eligible answer whether the study design is spatially properly replicated (see the cited reference above) or not. Based on the description in the manuscript I assumed that the study is based on a sampling scheme implemented in one spatial unit, thus the generalization of the result are biased due to lack of true spatial replicates. In other words, there is no evidence that the differences in species responses to the studied spatial configuration are true differences or just the result by the chance.
3. The author do not know own applied methods and results. Based on the author replies for the revision, they do not know what kind of methods they used for the data analysis. They consistently mix the general linear model and the non-linear mixed model. There is no standard protocol reported of data analysis, some arbitrarily condition were set without any references or justification (ie. estimation of the sampled spatial unit by a pitfall trap).
Author Response
Reviewer 2 (round 2)
I have received the 1st revision from the suggested manuscript. I’m regret to report that the quality of the manuscript do not meet the criteria at the Journal Agriculture. My suggestion is based on the following reason below:
- Author cannot provide eligible documentation of the changes in the text and figures during the revision. The replies are rather short and general there is no clear professional/scientific reasoning therein. Some of the replies are under the level of the expected quality of the language that supposed to be used in scientific communication. This issues supposed to be controlled by the author firmly. The author did not thoroughly revised the manuscript, some major issues have replied, some of them are ignored without any reason.
I looked once more in my former answers to reviewer 2. I cannot find any comment without answer, but several in which I do not agree with the reviewer, e.g (1) spillover. I also explained the calculation of weighted mean. It is a usual and common statistic tool in calculating such values used in the present study.
- The author does not know the concept of true spatial replicates in ecological studies to avoid pseudoreplication (Hurlbert, Stuart H. “Pseudoreplication and the Design of Ecological Field Experiments.” Ecological Monographs 54, no. 2 (1984): 187–211. https://doi.org/10.2307/1942661.). In addition, the author was not able to provide eligible answer whether the study design is spatially properly replicated (see the cited reference above) or not. Based on the description in the manuscript I assumed that the study is based on a sampling scheme implemented in one spatial unit, thus the generalization of the result are biased due to lack of true spatial replicates. In other words, there is no evidence that the differences in species responses to the studied spatial configuration are true differences or just the result by the chance.
Dear reviewer 2
I regret the misunderstanding concerning pseudoreplication. I hoped that adding of the grid map of pitfall traps helped you to solve your problems with pseudoreplication. I agree with you that using only one study area can be considered as pseudoreplication and we have discussed this problem in detail in our working group. But it is not realistic to assume that a financial support will be given for the investigation of several landscape sections as in the present study. The groups used here are certainly not pseudoreplicate as you can see from the map. It is true that the overall climate conditions are the same in all field parts. However, individual traps differ in the agricultural fruits (see method), partly in the pattern of soil conditions (loamy sites versus sandier sites), inclination (south slope versus shady parts near the forest). Moreover, although climate is the same all over the fields, microclimate differ according to insolation (caused by differing inclination and cover by different fruits). Nevertheless, the problem using one landscape section exists. I analysed this problem in an earlier paper and, therefore, added a sentence describing the results:
Line 59: A comparison of the investigated Ritzerau fields with 53 other field sites in Schleswig-Holstein showed that the overall differences of carabid assemblages is weak independent of farming practices. Nevertheless, fields can be separated in loamy and extreme sandy sites. The investigated Ritzerau fields belong to the group on loamy sites, which is most common in Schleswig-Holstein. Thus, the investigated landscape can be considered as representative for most arable field situations in Schleswig-Holstein [7].
3. The author do not know own applied methods and results. Based on the author replies for the revision, they do not know what kind of methods they used for the data analysis. They consistently mix the general linear model and the non-linear mixed model. There is no standard protocol reported of data analysis, some arbitrarily condition were set without any references or justification (ie. estimation of the sampled spatial unit by a pitfall trap).
Dear reviewer
It is really difficult to understand what you want to hear. What do you mean with “the author do not know own applied methods and results.” It would be very helpful to give concrete hints how you want to change the text. It is difficult to answer on such general criticism. What do you mean with no justification of “ estimation of spatial unit by a pitfall trap.” The calculation is described in line 110 – 130. Where are the problems? I have the number of pitfall traps and the space of the field area. Both values are given by the landscape structure. This calculation is very simple and certainly do not need a reference.